# Vitiligo Treated with Combined Piperine-Based Topical Treatment and Narrowband Ultraviolet B Therapy: Follow-Up with Reflectance Confocal Microscopy

**DOI:** 10.3390/diagnostics14050494

**Published:** 2024-02-25

**Authors:** Cristina Bertoli, Johanna Chester, Chiara Cortelazzi, Silvana Ciardo, Marco Manfredini, Sergio Di Nuzzo, Shaniko Kaleci, Giovanni Pellacani, Francesca Farnetani

**Affiliations:** 1Department of Dermatology, University of Modena and Reggio Emilia, 41123 Modena, Italy; 177733@studenti.unimore.it (C.B.); shaniko.kaleci@gmail.com (S.K.); 2Department of Medicine and Surgery, University of Parma, 43125 Parma, Italy; 3Dermatology Clinic, Department of Clinical Internal, Anesthesiological and Cardiovascular Sciences, Sapienza University of Rome, 00196 Rome, Italy

**Keywords:** vitiligo, confocal laser microscopy, piperine, cream, topical

## Abstract

Background: Reflectance confocal microscopy (RCM) has a defined in vivo morphology of vitiligo and re-pigmentation. Combination therapies seem more effective than monotherapies. Objective: We aim to describe the clinical and RCM features of re-pigmentation with combined narrowband ultraviolet B (NB-UVB) and piperine-based topical treatment in localized vitiligo. Methods: Eight patients enrolled at a single center received combined treatment: topical treatment was applied twice daily + NB-UVB twice weekly for 2 × 2-month periods. Clinical changes were analyzed by the Vitiligo Noticeability Scale (VNS) and percentage of re-pigmentation. The evaluator agreement was assessed. Predefined RCM features had the presence/absence of (i) blood vessels, (ii) dendritic cells, and the quantity of (i) an irregular honeycombed pattern and (ii) non-pigmented papillae. Clinical and RCM monitoring was performed at the baseline, 2, 3, 5, and 7 months. Results: Macules were “slightly less noticeable” with 25–50% re-pigmentation. Irregular honeycomb patterns and non-pigmented papillae were significantly less frequently observed, and in less extended areas (T1 vs. T2, *p* = 0.039; T0 vs. T1, *p* = 0.005 and T2 vs. T4, *p* = 0.033). Dendritic cells and blood vessels improved, with significant changes in blood vessels (T1 vs. T2, *p* = 0.005 and T3 vs. T4, *p* = 0.008). Conclusions: RCM confirms the morphological changes induced by combined treatment for localized vitiligo.

## 1. Introduction

Vitiligo is an acquired autoimmune skin disease associated with a genetic predisposition. Metabolic action, oxidative stress, cell detachment abnormalities, and environmental factors have all been implicated in the destruction of functional epidermal melanocytes [1,2,3]. Vitiligo is the most common cause of cutaneous depigmentation, with an estimated prevalence of 0.5–2.0% of the world’s population, with onset usually between 10 and 30 years of age [2]. Cosmetic unease is often referred to by patients as vitiligo commonly involves aesthetically delicate skin [4]. Vitiligo patients can suffer associated depression and anxiety, low self-esteem, and social isolation [4]. The clinical diagnosis of vitiligo is usually uncomplicated [2]. Observed by the naked eye, it appears as sharply scattered, demarcated, achromic macules, and according to its localization and extent, vitiligo is either classified as generalized or localized [2].

As a multifactorial disorder, vitiligo’s precise etiology and pathophysiology are complex. There is a lively debate about the various theories regarding the loss of melanocyte function [2]. Genetics, autoimmunity, oxidative stress, and neurological system dysfunction have been associated with vitiligo, but vitiligo phenotypes cannot be explained by one of these mechanisms alone [2]. The most probable explanation is that disparate mechanisms contribute to the same clinical result [5]. This convergence theory, which combines all existing theories into a comprehensive one, suggests that several mechanisms contribute to the reduction in melanocyte variability [5]. Despite each of these pathogenetic hypotheses being under continual discussion, there is now an agreement on the autoimmune and oxidative stress theories as the leading processes in vitiligo pathogenesis [6]. This pathogenic aspect is an important acknowledgment because the therapeutic rationale for using nb-UVB lamps for vitiligo is precisely at the basis of the oxidative cause of melanocyte dysfunction. 

Schallreuter et al. [7] and Telegina et al. [8] established the pterin/H_2_O_2_-related origin of vitiligo, which is effectively cured with narrowband UVB. 

Currently, the only topical treatment approved for vitiligo by the U.S. Food and Drug Administration is ruxolitinib [9]. Despite approval, the availability of ruxolitinib is limited, and real-life data are still lacking. Current clinical guidelines recommend topical corticosteroids (TCSs), topical tracolimus, narrowband ultraviolet B (NB-UVB), and combination therapies [10,11]. The Vitiligo Working Group (VWG) recommendations include the use of NB-UVB for vitiligo, and they specify a dosing protocol, frequency of administration, and a follow-up schedule [12]. 

As is the case with other pigmentary disorders, studies dedicated to the effectiveness of plant-based therapies (including psoralens, flavonoids, polyphenols, glycosides, etc.) have been undertaken. However, specific components and mechanisms of action are considered unclear [13]. Some natural drugs, such as baicalein, vitexin, and maclurin, have proven to inhibit the damaging effect of H_2_O_2_-induced oxidative stress on melanocytes. Studies have reported the counterstaining of depigmented mouse skin lesions, but strong evidence that natural products can prevent or treat vitiligo is still lacking [13]. As an auxiliary means of phototherapy, plant-derived compounds with antioxidant properties are becoming an attractive choice for the treatment of vitiligo [14].

Piperine is an alkaloid-based extract with many physiological effects, including antioxidant and anti-inflammatory actions [15]. Piperine was proven in a mouse model to stimulate melanocyte proliferation and dendrite formation and has an enhanced effectiveness when associated with UV [16]. As demonstrated by Thomas et al. [17], the combination of NB-UVB and potent TCS is likely to be superior to potent TCS alone. Few studies have explored NB-UVB/topical piperine combination therapy in humans, but preliminary results in animals suggest that combination therapy is superior to NB-UVB treatment alone [18].

Treatment effectiveness in vitiligo trials is often evaluated by a simple “percentage of re-pigmentation” scale (0–100) [19]. Critics emphasize that this approach does not consider the subjective interpretation by the dermatologist and/or patient of the treatment undertaken. Further, most vitiligo studies do not include histopathological data of cellular response due to the inherent limitations of biopsy requirements. 

Reflectance confocal microscopy (RCM) is a non-invasive tool that offers real-time skin imaging at a nearly cellular histological resolution [20,21], enabling treatment monitoring by the visualization of re-pigmentation without the need to perform a biopsy. Previous studies have reported RCM features of vitiligo and re-pigmentation areas prior to and following therapy [22,23,24]. Vitiligo is characterized by an abundance of irregular honeycomb patterns and non-pigmented papillae and the absence of dendritic cells and blood vessels.

Ardigo et al. [22] described the presentation of vitiligo with RCM via the absence of pigmented keratinocytes (seen as irregular honeycombed), with limited weak, bright cells and slight inflammation in the epidermis and complete or partial loss of pigmented rings (seen as non-pigmented papillae) (also criteria for early vitiligo diagnosis) with rare or absent melanocytes in the dermal-epidermal junction (DEJ). Following treatment, Ardigo et al. described the observation of re-pigmentation by the appearance of large dendritic or irregularly distributed round, bright cells, pigmented keratinocytes, and a cobblestone pattern in the epidermis, and pigment around the hair follicle and the active proliferation of melanocytes at the DEJ [22]. 

We aim to describe the clinical presentation and common RCM features associated with pigmentation captured over the study period of treatment with combined topical piperine and NB-UVB treatment in selected patients with localized vitiligo. 

## 2. Materials and Methods

### 2.1. Study Design, Patient Selection and Treatment Protocol

This study is a preliminary, single-center, observational study of patients with non-facial, localized vitiligo screened at the Department of Dermatology, Policlinico University Hospital, Modena, Italy. Consenting patients were enrolled between March 2021 and September 2022. 

The study criteria further specified the inclusion of subjects with ≥Fitzpatrick skin type II. The criteria specified the study’s exclusion of patients with (i) previous cutaneous neoplasms or (ii) the current assumption of photosensitive medications. Consenting patients were prescribed the experimental use of piperine treatment in combination with NB-UVB, according to a pre-determined study protocol. This study was approved by the local Ethics Committee (Prot # 583/2019), and all participants gave written informed consent.

As per the protocol, the application of topical piperine cream (Cromovit forte, Pharcos, Florence, Italy) was prescribed twice daily for 7 months. NB-UVB, a TL-01 lamp of a phosphor-coated fluorescent bulb emitting radiation wavelengths between 310 and 315 nm, was performed twice weekly during months 1, 2, 4, and 5 from treatment initiation (4 months in total). To avoid phototoxic adverse effects, a fixed dosing protocol was initiated at 200 mJ/cm^2^, which was incremented by 10–20% per session up to a maximum of 3000 mJ/cm^2^ for the body. In cases of erythema, the dosage was modified according to intensity (as per color) and symptomatology. In the case of pink, asymptomatic erythema lasting >24 h, the dose was maintained until resolution, followed by an increase of 10–20%. In the case of bright red, asymptomatic erythema, radiation was suspended until the area healed (returned to a light pink color), and dosage was reinstated to the last tolerated dose. Patients were advised to avoid sun exposure during therapy duration, and sunscreen was applied to patients with skin photo type I-III during the individual phototherapy sessions [12]. For phototherapy on the face or close to the genitalia, masks were used. 

Clinical evaluations and RCM monitoring were performed at the baseline (T0), 2 (T1), 3 (T2), 5 (T3), and 7 (T4) months. Figure 1 outlines the study protocol timeline.

Clinical image evaluations

Each clinical image acquired during the study period was blindly matched with baseline images, and three dermatologists (senior expert, expert, and resident in training) were asked to evaluate the follow-up images (Figure 2b) compared to the baseline image (Figure 2a) according to the following: The presence or absence of vessels in the chalky-white target macule area;The Vitiligo Noticeability Scale (VNS): (1) more- (2) as- (3) slightly less- (4) less-, and (5) no longer-, noticeable [12];The percentage of re-pigmentation: (1) 0–24% (2) 25–49% (3) 50–74% (4) 75–100%.

### 2.2. RCM Imaging

RCM images were captured with Vivascope 1500^®^; MAVIG GmbH, Munich, Germany. RCM instrumental and acquisition methods were previously described elsewhere [13]. To ensure the capture of RCM images in precisely the same location throughout the study period, an area was designated on a piece of transparent paper for each patient. Briefly, for each macule, a complete set of ≥3 Viva-Block mosaics (epidermal layers, the dermo-epidermal junction [DEJ], and the upper dermis) was captured. RCM features of interest were predefined, according to those identified by Ciardo et al., as both diagnostic and ideal for skin response: the presence/absence of (i) blood vessels, (ii) dendritic cells, and the quantity of (i) an irregular honeycombed pattern and (ii) pigmented papillae [25]. Follow-up/post-treatment investigations were performed via telephone with the patients at 10 and 12 months from treatment initiation. 

### 2.3. Treatment Efficacy

Treatment efficacy was assessed retrospectively by qualitative and quantitative evaluations of clinical and RCM images (Figure 2). RCM images were blinded and randomly presented, and a dermatologist and instrumental RCM technician were invited to individually evaluate images for the presence/absence or the amount of selected RCM common pigmentation features. Feature amounts were considered as the observation of a specific feature in a percentage of the mosaic image (0, <25%, 25–50%, and 50–75% >75%) [19].

### 2.4. Adverse Events

Any adverse events or reactions reported by patients or observed by the physicians were recorded at each study point during the study period. 

### 2.5. Statistical Methods

Statistical analysis was performed using STATA^®^ software version 17 (StataCorp. 2021. Stata Statistical Software: Release 17. College Station, TX, USA: StataCorp LLC). Descriptive statistics were presented for baseline demographic clinical characteristics for the entire group. Continuous variables were presented as the number of patients (*n*), the mean, standard deviation (SD), minimum (min), and maximum (max) and compared between subgroups using Student’s paired *t*-test; analysis of variance (ANOVA) was used to evaluate the differences in the parameters under examination for variables with three or more categories while categorical variables were presented as the frequency (*n*, percentage [%]) and compared using Pearson’s chi-squared test. Moreover, κ was also calculated in the evaluation of the agreement between the dermatologists. We selected κ-weighted statistics as the measure of agreement because our variable of interest was not binary [26,27]. Kappa is a measure of this difference, standardized to lie on a −1 to 1 scale, where 1 is a perfect agreement, 0 is exactly what would be expected by chance, and negative values indicate an agreement less than chance, i.e., potential systematic disagreement between the observers. The interpretation of the agreement adopted here is less than chance agreement (κ < 0), slight agreement (κ = 0.01 to 0.20), fair agreement (κ = 0.21 to 0.40), moderate agreement (κ = 0.41 to 0.60), substantial agreement (κ = 0.61 to 0.80), and almost perfect agreement (κ = 0.81 to 0.99). The interpretation of reproducibility adopted is marginal (κ = 0.00 to 0.40), good (κ = 0.40 to 0.75), and excellent (κ > 0.75) [28]. Margin statistics were used to obtain the predicted probability of a previously fit model at fixed values of VNS among the baseline and time of treatment. A *p* < 0.05 was considered statistically significant.

## 3. Results

We initially enrolled 9 participants, with 1 immediate dropout (data were not included in the analyses). The cohort included a 5/3 male/female ratio and the average patient age was almost 49 years old (33–69). Just over half of the target vitiligo patches were located on the trunk (*n* = 5), and most patients were classified as phototype II (*n* = 6). The number of comorbidities was reported at treatment initiation; see Table 1. One patient dropped out following the 2-month visit due to a novel disease appearance (a solid organ tumor), and the subsequent impossibility of adhering to the study protocol. Seven patients completed the protocol with complete sets of clinical and RCM images available for assessment.

### 3.1. Clinical Evaluation

Evaluators’ observations at the baseline revealed the presence of visible vessels in only 1/8 macules, and by the study’s end, vessels were observed in 5/7 macules; see Table 2.

The mean scores of both the VNS and the percentage of re-pigmentation scales according to each evaluator proved that, throughout the study period, patients’ vitiligo was less noticeable and with larger areas of pigmentation compared to the baseline; see Table 3, Figure 3. 

Evaluator agreement

The mean levels of agreement between evaluator 1, evaluator 2, and evaluator 3 were fair (κ = 0.22 and κ = 0.32), whilst between evaluators 2 and 3, the level of agreement was substantial (κ = 0.67), see Table 4.

### 3.2. RCM Evaluation

At the baseline, the epidermal layer was characterized by the irregular honeycomb pattern observed in all macules (25–50% [*n* = 3] and 50–75% [*n* = 5]). By the treatment end, the irregular honeycomb pattern was not observed in 3 macules and in less extended areas of the mosaic images in the remaining 4 macules (<25% [*n* = 3], 25–50% [*n* = 1]). A significant difference was observed between evaluations at 2 and 3 months (*p* = 0.039).

A similar pattern of observation was noted for non-pigmented papillae at the dermo-epidermal junction (DEJ). Non-pigmented papillae were observed more heavily at the baseline (50–75% [*n* = 1] and >75% [*n* = 7]) compared to the treatment end (<25% [*n* = 4], 25–50 [*n* = 2], 50–75% [*n* = 1]). Significant differences were observed between the baseline and 2 months (*p* = 0.005) and again between 3 and 7 months (*p* = 0.033).

The presence of dendritic cells and blood vessels fluctuated during the study period. At the baseline, dendritic cells were not observed in any macule, and blood vessels were evident in one macule only. During the study period, dendritic cells were observed in half of the macules (*n* = 4) at 3 months and one macule at the study’s end. Blood vessels were observed in almost all of the macules at 3 months (*n* = 7) and in five macules at the study’s end. Significant differences in blood vessels were observed between 2 and 3 months (*p* = 0.005) and between 5 and 7 months (*p* = 0.008); see Table 2.

### 3.3. Adverse Events

No adverse events or reactions were reported during the study period by the patients or observed by the clinicians.

## 4. Discussion

Clinically enrolled participants with vitiligo were mainly characterized by the absence of blood vessels in a chalky-white skin macule. Following NB-UVB treatment and a combined topical piperine-based therapy, blood vessels were reinstalled in almost all macules, and the VNS and percentage of re-pigmentation scales were improved. In vivo molecular analysis with RCM proved that the regularity of the epidermal honeycomb pattern improved, the concentration of non-pigmented papillae decreased, and the number of evident dendritic cells and blood vessels increased.

Vitiligo is a common skin disorder due to complex pathogenesis. Briefly, in patients with vitiligo, melanocytes are susceptible to oxidative stress, where environmental stressors cause alterations in the antioxidant system. Oxidative stress leads to further oxidative stress and cell damage, which eventually causes inflammation through a positive feedback loop [2]. Piperine is the bioactive alkaloid ingredient of black pepper (Piper nigrum) and has numerous physiological effects, including antioxidant and anti-inflammatory actions [29]. Therefore, the choice of an agent that has an antioxidant and anti-inflammatory action, combined with NB-UVB, warrants further research in vitiligo therapy. Furthermore, piperine-based topical treatment may be a valid and safer alternative for ongoing treatment than prolonged topical corticosteroids [18]. Our results suggest that the application of a piperine-based topical treatment combined with NB-UVB treatment increases the clinical and morphological effect of the re-pigmentation of the vitiligo macule without any adverse events. Further studies are necessary to confirm our preliminary results.

Although our study does not include a control group, there have been numerous studies that have reported improved results associated with combined treatments compared to monotherapies, including NB-UVB alone. Batchelor et al. performed a three-arm study where patients were randomly allocated to active topical corticosteroids + dummy NB-UVB, active NB-UVB + placebo ointment, or active topical corticosteroids ointment + active NB-UVB. Patients were provided with a personal, hand-held NB-UVB to be used over the study period. The authors reported significant comparative improvements in participants treated with combined therapy [30]. Shafiee et al. studied the use of piperine treatment with NB-UVB and compared them to an NB-UVB-only control group. NB-UVB was performed every other day for 3 months. The authors reported a significant difference in clinical re-pigmentation throughout the study period in favor of the combined therapy approach [18].

Clinically, target macules were assessed by the evaluators as “slightly less noticeable”, with <50% of the lesion re-pigmented. Interestingly, the effect of the treatment, as visualized with RCM, seemed to induce a constantly increasing improvement in the irregular honeycomb pattern and non-pigmented papillae towards the end of the study period (T4), whereas the peak response for dendritic cells and vessels was observed between T2 and T3, during/after the second NB-UVB treatment period. Ardigo et al. were the first to describe and compare the RCM features of macules and non-lesional skin following NB-UVB treatment in patients with vitiligo and a control group of “normal skin”. Re-pigmented skin was characterized by a regular honeycombed pattern and large, dendritic, or round-to-oval bright cells irregularly distributed at the dermal-epidermal junction [22]. The authors hypothesize that the fluctuation in the presence of dendritic cells and vessels observed with RCM in our study could be due to the protocol of NB-UVB proposed. Batchelor et al. noted that participants who adhered to ≥75% of expected treatments were more likely to achieve treatment success in the combination group compared with topical corticosteroids [30]. The results provided by Shafiee et al. include observations throughout the treatment period, but evidence of ongoing treatment efficacy was not reported [18].

The light source used for NB-UVB phototherapy in our study has a peak emission at 311 nm. We maintained an emission range between 310 and 315 nm to minimize superfluous radiation, thereby reducing the risk of severe burning or other UV radiation-associated cutaneous side effects [12]. Our protocol also specified twice-weekly NB-UVB sessions over an interrupted 4-month period (over a complete 7-month follow-up period). A twice weekly session regime was established to minimize the impact of therapy commitment on the patient’s lifestyle. Recommendations from the VWG highlight that re-pigmentation is dependent on the total number of phototherapy sessions, with earlier re-pigmentation observed in patients undergoing three weekly sessions. However, there is currently no evidence directly comparing any efficacy difference between the regimes of sessions applied twice or three times per week [12].

Our study is not without limitations. The cohort is small and does not include a control group of matched participants with “normally pigmented skin” or the investigation of the non-lesional skin of enrolled participants. Further, the study RCM follow-up period to assess treatment efficacy was restricted to 2 months post-intervention.

## 5. Conclusions

The non-invasive, morphological RCM assessment of combined piperine-based topical treatment and NB-UVB affirms that morphological changes are induced by treatment. Further studies are required to confirm these preliminary results. The use of RCM provides evidence of ongoing morphological changes during the treatment period.

## Figures and Tables

**Figure 1 diagnostics-14-00494-f001:**
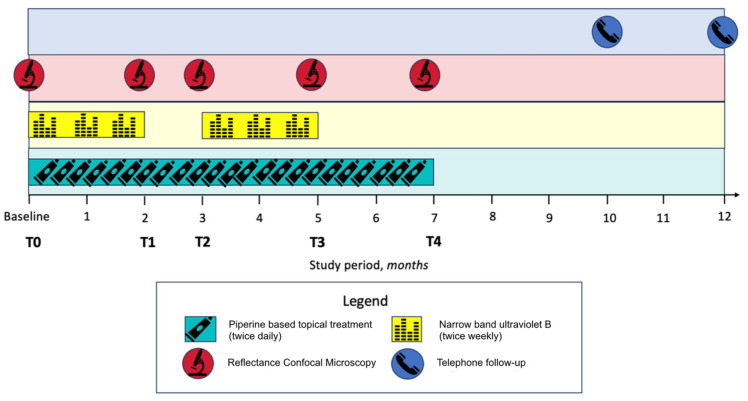
Study protocol timeline.

**Figure 2 diagnostics-14-00494-f002:**
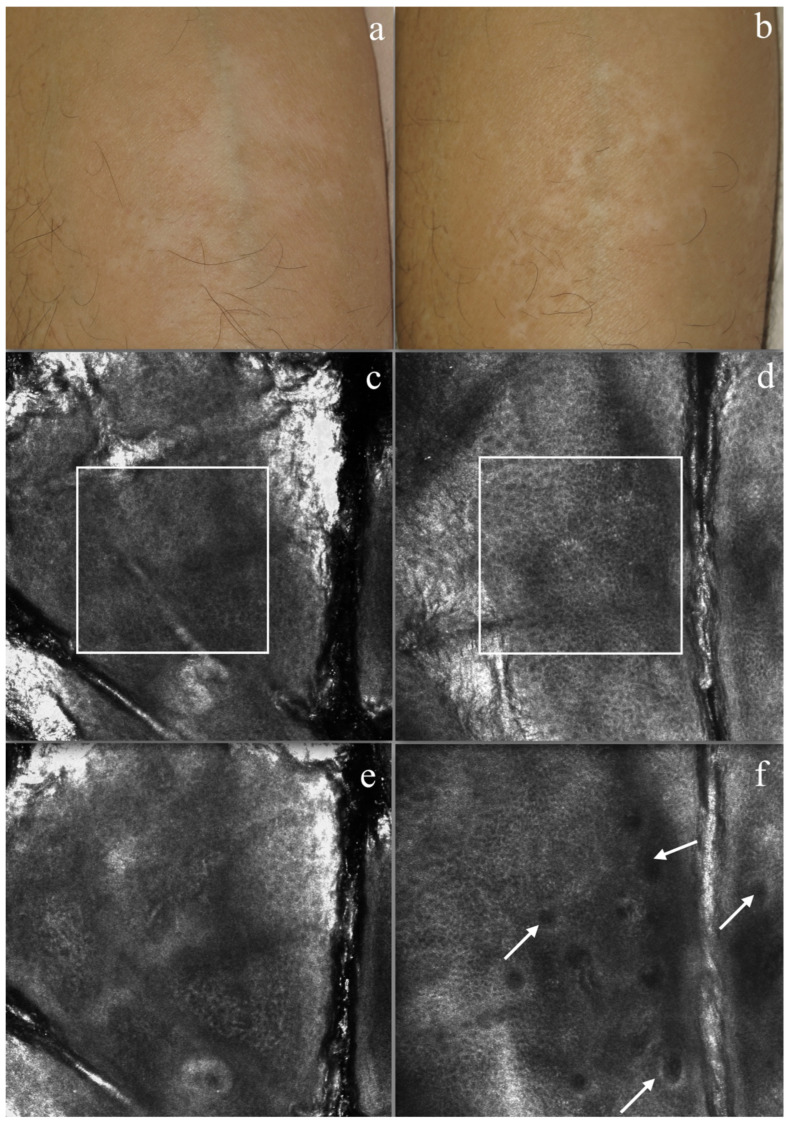
Vitiligo patient. Clinical picture of forearm (**a**) before therapy (**b**) and after treatment (T4). Reflectance confocal microscopy (RCM) imaging: (**c**) no bright structures were detected in the upper epidermal layers, with an absence of a honeycombing pattern and pigmented keratinocytes absent or undefined. (**d**) At the epidermal layers, after the treatment re-pigmentation of keratinocytes and the presence of honeycombing, numerous round cells were seen. (**e**) At DJE, none of the bright dermal papillary rings were normally seen at the demo epidermal junction. (**f**) At DJE after treatment, some dermal papillary rings could be defined (white arrows).

**Figure 3 diagnostics-14-00494-f003:**
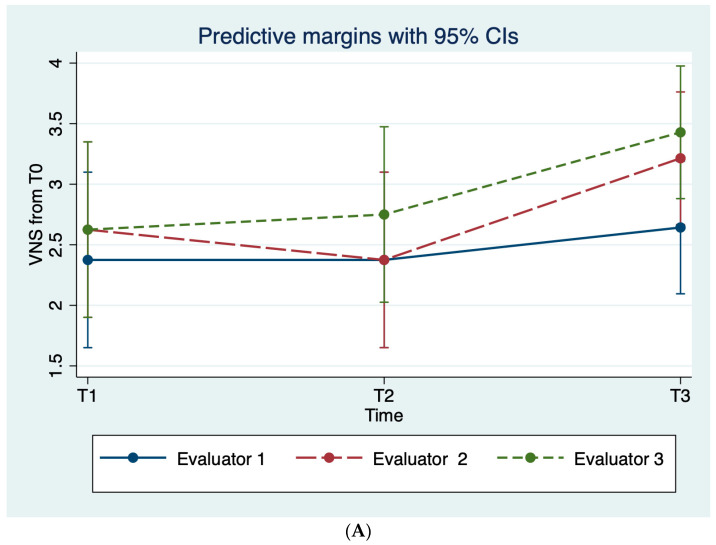
Average evaluator scoring on the (**A**) VNS scale and (**B**) the percentage of re-pigmentation throughout the study period.

**Table 1 diagnostics-14-00494-t001:** Patient characteristics.

Characteristic, *n (%)*	Total (*n* = 8)
Female	3 (37.5)
Age, *mean ± SD (range)*	48.8 ± 11.7 (33–69)
Macule location	
Upper/lower limb	3 (37.5)
Trunk	5 (62.5)
Fitzpatrick skin phototype	
II	5 (62.5)
III	2 (25.0)
IV	1 (12.5)

**Table 2 diagnostics-14-00494-t002:** Selected clinical and reflectance confocal microscopy features observed throughout the study period.

Patterns/Features		T0 *n* = 8	T1 *n* = 8	T2 *n* = 8	T3 *n* = 7	T4 *n* = 7
Clinical						
Chalky white target macule area	No vessels	7 (87.5)	1 (12.5)	1 (12.5)	2 (28.6)	2 (28.6)
	vessels	1 (12.5)	7 (87.5)	7 (87.5)	5 (71.4)	5 (71.4)
RCM						
Irregular honeycombed pattern *	Absent	0 (0)	0 (0)	2 (25)	3 (37.5)	3 (37.5)
	<25%	0 (0)	1 (12.5)	2 (25)	2 (25)	3 (37.5)
	25–50%	3 (37.5)	4 (50)	4 (50)	2 (25)	1 (12.5)
	>50–75%	5 (62.5)	3 (37.5)	0 (0)	0 (0)	0 (0)
Non-pigmented papillae ^°	<25%	0 (0)	0 (0)	0 (0)	0 (0)	4 (50)
	25–50%	0 (0)	1 (12.5)	3 (37.5)	5 (62.5)	2 (25)
	>50–75%	1 (12.5)	0 (0)	3 (37.5)	2 (25)	1 (12.5)
	>75–100%	7 (87.5)	7 (87.5)	2 (25)	0 (0)	0 (0)
Dendritic cells	Absent	8 (100)	7 (87.5)	4 (50)	5 (62.5)	6 (75)
	Present	0 (0)	1 (12.5)	4 (50)	2 (25)	1 (12.5)
Vessels ^#$^	Absent	7 (87.5)	1 (12.5)	1 (12.5)	2 (25)	2 (25)
	Present	1 (12.5)	7 (87.5)	7 (87.5)	5 (62.5)	5 (62.5)

RCM, Reflectance confocal microscopy. ^ T0 vs. T1 (*p* = 0.005); ^#^ T1 vs. T2 (*p* = 0.005); * T2 vs. T3 (*p* = 0.039); ° T2 vs. T4 (*p* = 0.033); and ^$^ T3 vs. T4 (*p* = 0.008).

**Table 3 diagnostics-14-00494-t003:** Mean changes reported from evaluators by the Vitiligo Noticeability Scale (VNS) and the percentage of re-pigmentation at each study time point.

		Evaluators		
Value, *Mean ± SD (Range)*		1	2	3
VNS				
T0–T1		2.37 ± 0.5 (2–3)	2.62 ±1.5 (1–5)	2.62 ± 0.7 (2–4)
T0–T2		2.37 ± 0.5 (2–3)	2.37 ±1.8 (1–4)	2.75 ± 0.8 (2–4)
T0–T3		2.28 ± 0.9 (1–4)	3.14 ±1.2 (2–5)	3.42 ± 0.9 (2–5)
T0–T4		3.0 ± 0.8 (2–4)	3.28 ±1.4 (1–5)	3.42 ± 0.9 (2–5)
Percentage of re-pigmentation				
T0–T1		1.12 ± 0.3 (1–2)	2.25 ± 1.2 (1–4)	1.85 ± 0.9 (1–4)
T0–T2		1.12 ± 0.3 (1–2)	2.0 ± 1.0 (1–4)	2.0 ± 1.9 (1–4)
T0–T3		1.42 ± 0.7 (1–3)	2.57 ± 1.1 (1–4)	2.57 ± 1.2 (1–4)
T0–T4		1.71 ± 0.9 (1–3)	2.71 ± 1.1 (1–4)	2.71 ± 1.3 (1–4)

VNS: 1, more noticeable; 2, as noticeable; 3, slightly less noticeable; 4, less noticeable; and 5, no longer noticeable. Percentage of re-pigmentation: 1, 0–24%; 2, 25–49%; 3, 50–74%; and 4, 75–100%.

**Table 4 diagnostics-14-00494-t004:** Evaluator agreement at each study time point.

	Evaluators		
Study Time Points, *κ*	1 vs. 2	1 vs. 3	2 vs. 3
T0–T1	0.2258	0.6000	0.4286
	0.0769	0.1304	0.6842
T0–T2	0.2258	0.5000	0.6667
	0.0968	0.1250	0.7838
T0–T3	0.3226	0.1765	0.7586
	0.1765	0.1714	0.5484
T0–T4	0.3438	0.5532	0.6769
	0.2687	0.3288	0.7812
Overall mean agreement	0.2171	0.3232	0.6661

## Data Availability

Data will be made available upon any reasonable request made to the corresponding author.

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
