# Peer review of "Vitiligo Treated with Combined Piperine-Based Topical Treatment and Narrowband Ultraviolet B Therapy: Follow-Up with Reflectance Confocal Microscopy"

_diagnostics, 2024, doi:10.3390/diagnostics14050494_

Round 1

Reviewer 1 Report

Comments and Suggestions for Authors

Review to „Vitiligo treated with combined piperine based topical treatment and narrowband ultraviolet B therapy: follow-up with reflectance confocal microscopy” by Bertoli et al.

The authors around Bertoli et al. show data of a small one-arm human study treating 8 vitiligo patients with a combination of topical treatment combined with UV-B. Analysis was done via VNS, percentage of re-pigmentation and predefined microscopy features. They do discuss obvious downsides of the study (small n, control, short time), but I still think it is worth as a communication. Paper reads well, I am happy to see a proper ethics description which is still not common these days. So overall, I believe with some minor modifications (see below) this can be a good small publication.

L153 “The cohort included 5 males“ while in tab.1 you only write females: 3 which is technically correct, it feels however strange. Rather state consistently a 5/3 m/f ratio

Comments on the Quality of English Language

Fig. 1 spelling error, also in small letters “confocal”

Fig. 1 “ultraviolet” (for consistent writing style)

Fig. 2 – scale bar missing (including in the legend)

Fig. 3 – I am sure there is an English word for valutatore ;-)

Fig. 3 – I believe it should by increase of percentage on the y-axis

Fig. 3 – what happened to T4?

Table 2 – a small “n” at T4

Author Response

we have added data regarding the characteristics of the narrow band UV used in the study, we have followed the indications of the references and guidelines that we have already indicated but it is actually more correct to insert this part, thanks

Reviewer 2 Report

Comments and Suggestions for Authors

This study used reflectance confocal microscopy (RCM) to observe the morphological changes of piperine combined with NB-UVB in the treatment of localized vitiligo, which confirmed the effectiveness of the combined treatment from the perspective of non-invasive imaging, it has certain clinical significance.

However, this paper has several shortcomings:

1. The sample size of the subjects is too small, only 8 participants, and the source of sample size calculation is not mentioned in this paper, so the reliability of the research results cannot be verified. It is recommended to increase the number of study participants.

2. Is the index of "irregular honeycomb pattern" in the RCM evaluation correct? As far as I know, the honeycomb pattern of RCM in vitiligo has not been damaged. It is recommended to consult multiple literatures for verification. The method section does not specifically propose how to ensure that the positioning of each RCM detection is consistent.

3. Some sentences in the paper are incorrect, such as "Currently there are no specific drugs approved for vitiligo " in the Introduction, in fact, there are some drugs approved for the treatment of vitiligo.

    4. The format of references is not uniform.

Comments on the Quality of English Language

The quality of the English in this article is good, but some grammar and sentence expression still need to be improved.

Author Response

we took into consideration and explored pterin/H2O2-related origin of vitiligo which effectively reinforces the concept underlying our study

Reviewer 3 Report

Comments and Suggestions for Authors

Minor Revision

Reference number 17 missing.

Line 114: pigmented papillae17. what does 17 represent?

Figure 1 quality can be improved. 

Author Response

 we reviewed citations and references, adding those proposed by the editor and a further consideration of the confocal which underlines its importance, thanks

Reviewer 4 Report

Comments and Suggestions for Authors

Dear Sirs,

I have read the manuscript and recommend its minor revision.

Please, take into account my comments and suggestions in the file attached.

Author Response

we have explored the use of plant-based derivatives in the literature regarding pigmentation pathologies and added this part in the introduction, providing further data to support the use of a piperine-derived cream, thanks 

Round 2

Reviewer 2 Report

Comments and Suggestions for Authors

This study used reflectance confocal microscopy (RCM) to observe the morphological changes of piperine combined with NB-UVB in the treatment of localized vitiligo, which confirmed the effectiveness of the combined treatment from the perspective of non-invasive imaging, it has certain clinical significance.

However, this paper has several shortcomings:

1. The sample size of the subjects is too small, only 8 participants, and the source of sample size calculation is not mentioned in this paper, so the reliability of the research results cannot be verified.  It is recommended to increase the number of study participants.

2. Is the index of "irregular honeycomb pattern" in the RCM evaluation correct?  As far as I know, the honeycomb pattern of RCM in vitiligo has not been damaged.  It is recommended to consult multiple literatures for verification.  The method section does not specifically propose how to ensure that the positioning of each RCM detection is consistent.

Comments on the Quality of English Language

The English level of this paper is of high quality, and only a few areas need to be improved.